# Economic impacts of melting of the Antarctic Ice Sheet

Simon Dietz [1] ✉ & Felix Koninx [2]

Melting of the Antarctic Ice Sheet (AIS) could contribute metres to global sea level rise (SLR) in the long run. We couple models of AIS melting due to rising temperatures, SLR, and economic impacts of SLR on coastlines worldwide. We report SLR projections close to the latest literature. Coastal impacts of AIS melting are very heterogeneous: they are large as a share of GDP in one to two dozen countries, primarily Small Island Developing States. Costs can be reduced dramatically by economically efficient, proactive coastal planning: relative to a no adaptation scenario, optimal adaptation reduces total costs by roughly an order of magnitude. AIS melting increases the social cost of carbon by an expected 7% on low to medium emissions scenarios and with moderate discounting. There is a tail risk of very large increases in the social cost of carbon, particularly on a high emissions scenario.

The AIS is losing mass due to global warming, raising global sea levels[1]. The AIS holds freshwater equivalent to 58 m of global SLR, more than eight times that of the Greenland Ice Sheet (GIS). Its contribution to SLR is accelerating[2] and could constitute the largest source of SLR in the future.

In economic modelling, the AIS (like the GIS) is often a 'missing risk'[3–6]—a gap that should be filled in order to better quantify the social cost of carbon and economically optimal strategies to respond to climate change, principally reducing greenhouse gas emissions but also adaptation. Sometimes AIS melting is further conceptualised as a tipping element in the climate system[7,8] and, as such, the AIS has loosely inspired a number of economic studies into climate tipping points using stylised damage functions[9–12]. However, these studies are disconnected from the actual process of ice sheet melting and SLR, because they work with abstract damage functions depending on global mean temperature.

We are aware of two previous studies of the economic impacts of AIS melting. Reference 13 built a simple model of melting of the West Antarctic Ice Sheet (WAIS) and incorporated it in the DICE IAM. They used the framework of survival analysis, whereby WAIS melting is formulated as a discrete tipping event, triggered by a stochastic process that depends on global mean temperature. If the tipping event is triggered, the WAIS henceforth contributes a constant annual amount to global mean sea level over many hundreds of years. An earlier study of WAIS melting was done by[14], using stylised SLR scenarios fed into the FUND IAM.

A wider literature has studied the global economic impacts of SLR from all sources, thus implicitly including a contribution from AIS melting[15–19]. Recent advances include refs. 20 and 21, which build different, spatially detailed models of the economic costs of SLR globally, using probabilistic local SLR projections[22]. Reference 20 emphasises coastal adaptation, with the options of protecting coastlines or retreating proactively where these maximise net economic benefits. We use this model as part of our study.

In this paper, we study the economic impacts of SLR from AIS mass loss, which we colloquially refer to as melting. We build a reduced-form model of AIS melting (a.k.a. an emulator or reduced-complexity model) that draws on recent advances in the glaciology literature[23–27]. It is designed to reproduce the behaviour of much richer, higher-complexity modelling. We then couple this model of AIS melting to a model of the economic impacts of SLR, which enables us to estimate country-level costs broken down into different categories, and to do so under different assumptions about the economic efficiency of adaptation. Then we place the coupled model within a broader, modular Integrated Assessment Model (IAM) that allows the contribution of AIS melting to the overall social cost of carbon to be assessed. Thus, we aim to integrate recent progress in both glaciology and coastal impacts/economic modelling. Our approach is similar to Nordhaus' ground-breaking recent study of GIS melting[28], turning the focus to Antarctica. We build on the two previous studies of the economic impacts of AIS melting in three ways. First, we can leverage progress in glaciology to construct a more realistic, process-based

[1]London School of Economics and Political Science (LSE), London, UK. [2]Arup, Bristol, UK. ✉e-mail: s.dietz@lse.ac.uk

dynamical model of ice mass loss. Second, we can cover the whole of the AIS, not just the WAIS. Third, we can build on recent advances in coastal impacts modelling to study the cost of AIS melting at high spatial resolution and under different assumptions about adaptation.

## Results

### Primer on melting of the AIS

The contribution of ice sheets to SLR is often divided into two categories: (i) the surface mass balance (SMB) contribution, and (ii) dynamic contributions. SMB is the balance of surface mass accumulation (precipitation) and ablation (melting) on the ice sheet. Dynamic contributions come from the physical transportation of grounded ice into the ocean through glacier flow. Once afloat, this ice contributes to SLR through the displacement of water. Both SMB and dynamic losses/gains can be combined to describe the ice sheet's total mass balance.

SMB contributes only a small portion of the total mass balance of the AIS[2], circa 10%, due to low levels of precipitation and melting in very cold, dry conditions. This contrasts with Greenland, where SMB constitutes up to 68% of total mass balance[29]. Moreover, climate change that leads to a warmer, wetter Antarctica will likely enhance accumulation more than ablation, particularly in East Antarctica, leading to a small negative contribution of SMB to sea levels, at least for the first several degrees of global mean surface temperature increase relative to pre-industrial[24].

Dynamic contributions are more important on the AIS, and the primary driving force is basal melting of floating ice shelves[25]. While melting of ice shelves makes a negligible direct contribution to SLR, these ice shelves provide stability to the seaward flow of upstream, grounded ice through buttressing, and the loss of restraining ice shelves will cause the ice flow to speed up. Ice-shelf thinning has been shown to decrease this buttressing/back stress in numerical models[30], and this has been confirmed with observations, particularly in the Amundsen Sea region[31,32]. Unlike the GIS, portions of the AIS are grounded below sea level on reverse or retrograde slopes that slope upwards in the direction of flow. This means portions of the AIS are more prone to relatively rapid collapse via the Marine Ice Sheet

Instability (MISI) mechanism[33,34]. Although basal ice shelf melting is considered the most important dynamic process on Antarctica in the near term[25], other dynamic processes could contribute to SLR, notably ice shelf hydrofracturing and Marine Ice Cliff Instability (MICI)[26,35–37].

### Modelling approach in brief

Our modelling approach is summarised in Fig. 1 and described in detail in the Methods and Supplementary Information.

We build separate, reduced-form models of SMB and dynamic contributions. The former uses a simple, adjusted linear relationship between SMB and global mean surface temperature change, building on refs. 23,24. The latter uses the reduced-form model of ref. 25, which is designed to emulate basal ice shelf melting and the resulting contribution of the AIS to global mean SLR in 16 state-of-the-art ice sheet models (the LARMIP-2 models). We also test the sensitivity of our cost estimates to including the SLR projections of ref. 26, which incorporate hydrofracturing and MICI processes omitted by the LARMIP-2 models, as well as the SLR projections of the Antarctic BUttressing Model Intercomparison Project (ABUMIP)[27], which explores an unrealistic scenario of extreme melting of the AIS' floating ice shelves. We treat the latter as a worst-case scenario.

We couple these models to the Coastal Impact and Adaptation Model (CIAM) of ref. 20, which builds on the Dynamic Interactive Vulnerability Assessment (DIVA) database[38]. DIVA partitions the world's coastlines into 12,148 segments with homogeneous physical characteristics. To make coupling possible, we statistically downscale our global SLR projections to the segment level. At the segment level, CIAM then estimates adaptation costs (protection and proactive retreat) and residual damages (reactive retreat, inundation, wetland costs, and flood costs) depending on what is assumed about the planning/policy response. The key contribution of CIAM over and above DIVA is the potential to adapt optimally to SLR at the segment level, i.e., to minimise adaptation and residual damage costs over a rolling planning horizon. We run both optimal and no adaptation scenarios to bound possible costs. Segment-level costs can be aggregated to the national and then global level.

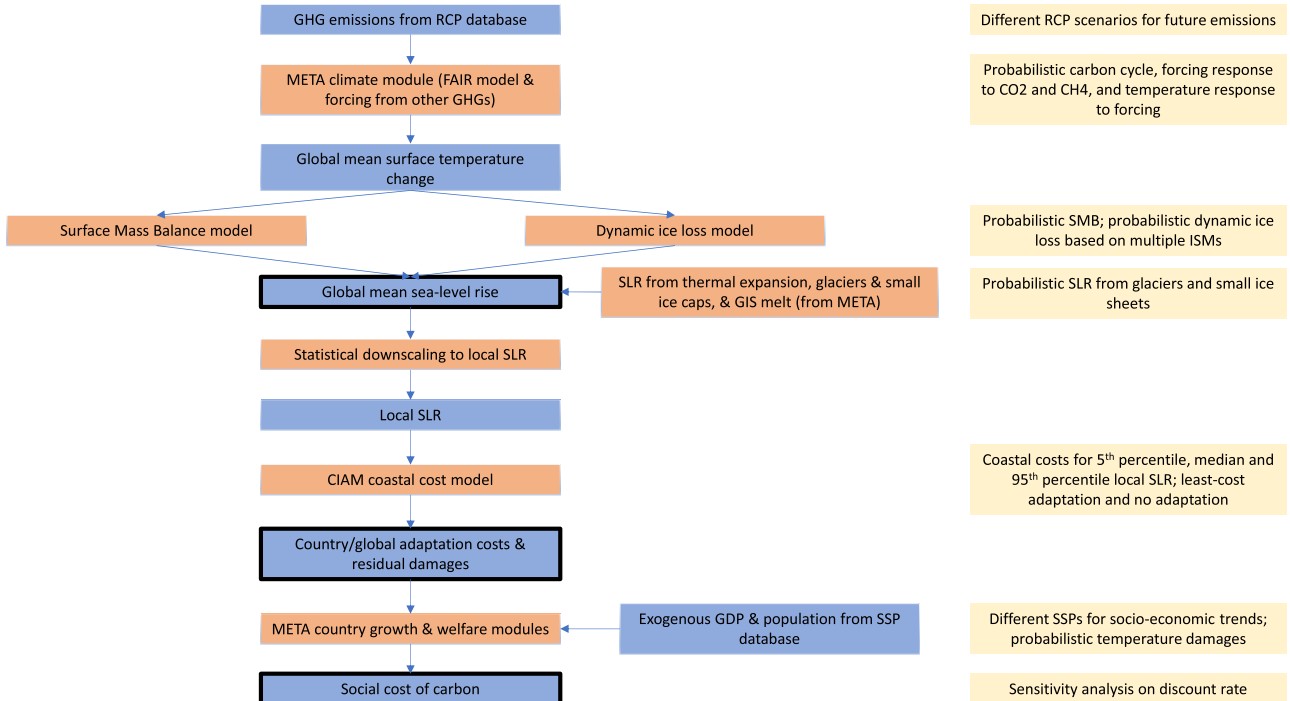

**Fig. 1 | Schematic diagram of model.** Blue boxes indicate input/output variables; thick outlines indicate output variables of main interest; orange boxes indicate modules; yellow boxes indicate key uncertainties considered.

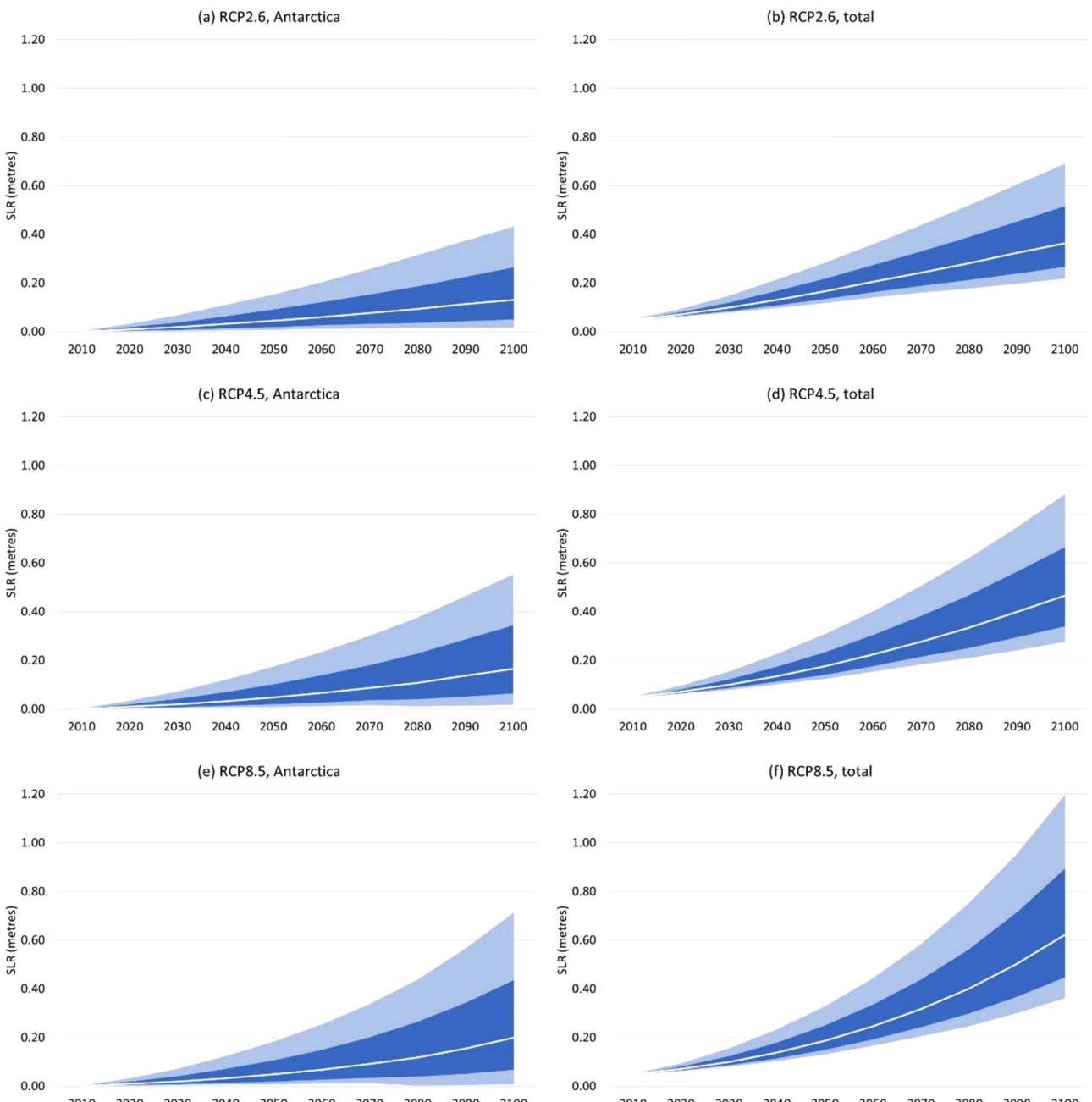

**Fig. 2 | Projection of Antarctica's SLR contribution and total SLR from all sources this century on RCP2.6, RCP4.5, and RCP8.5. a** RCP2.6, Antarctica's contribution only; **b** RCP2.6, all sources of SLR; **c** RCP4.5, Antarctica; **d** RCP4.5, all sources; **e** RCP8.5, Antarctica; **f** RCP8.5, all sources. Projections are relative to 2000. White line represents median value; dark shaded area represents the 67% confidence interval; light shaded area represents the 90% confidence interval. Results from a Monte Carlo simulation with sample size 50,000.

We then take the coupled AIS-CIAM model and place it, as a module, within the larger META (Model of Economic Tipping point Analysis) IAM[8]. This allows us to estimate the contribution of AIS melting to the social cost of carbon, that is, the present value of the economic costs of emitting an extra tonne of carbon dioxide, a key input to policy processes worldwide[39,40]. META contains separate modules representing thermal expansion of the oceans, melting of glaciers and small ice caps, and GIS melting, so we can model the incremental impact of AIS melting and account for possible interdependencies relating to coastlines and costs.

### Sea level rise
Figure 2 shows our estimates of global mean SLR. Our interest is primarily in SLR from AIS melting. For wider context, however, we also show global mean SLR from all sources, which is estimated by adding in SLR from thermal expansion, glaciers and small ice caps, and GIS melting, using the META model.

Focusing on the AIS contribution (panels a, c and e), we estimate median (mean) SLR of 0.16 m (0.21 m) in 2100 on the RCP4.5 scenario, within a 90% confidence interval of 0.02–0.55 m. These are naturally close to the estimates of ref. 25, whose model of basal ice shelf melting we replicate. The difference is primarily due to the small role of SMB, as well as slightly different global mean surface temperatures being inputted to the model (FAIR instead of MAGICC). On the low emissions RCP2.6 scenario, we project median (mean) SLR from AIS melting of 0.15 m (0.18 m) in 2100, within a 90% C.I. of 0.03–0.45 m. On the high emissions RCP8.5 scenario, we project median (mean) SLR of 0.20 m (0.26 m) in 2100, within a 90% C.I. of 0.01–0.71 m. Thus, higher

emissions/temperature scenarios spread the distribution of SLR in 2100, particularly in the upper tail. Figure SI6 disaggregates the AIS contribution to SLR on the RCP4.5 scenario into the contribution from SMB and the dynamic contributions from each of Antarctica's five regions. The SMB contribution is negative but small. The dynamic contributions are positive and several times larger, with the largest sources being the Weddell, East Antarctica, and Ross regions, in descending order.

We can compare our estimates with the recent synthesis in the IPCC 6th Assessment Report[29]. IPCC AR6 estimates median SLR from the AIS of 0.11 m in 2100 on RCP4.5 (67% C.I. of 0.03–0.29 m), also 0.11 m on RCP2.6 (67% C.I. of 0.03–0.27 m), and 0.12 on RCP8.5 (67% C.I. of 0.03–0.34 m). Thus, our median projections are somewhat higher and more sensitive to the emissions/temperature scenario, although still well within the 67% C.I. AR6 also provides a "low confidence" projection of the Antarctic contribution to SLR in 2100 on RCP8.5, based on structured expert judgement[41] and MICI[26]. Thus, it includes a broader range of uncertainties. The median of this projection is 0.19 m (67% C.I. of 0.02–0.56 m), which is very close to ours.

Regarding SLR from all sources (panels b, d and f), median SLR on RCP4.5 is 0.47 m in 2100, which is fairly close to the IPCC AR6 median estimate of 0.56 m and within their 67% C.I. of 0.44–0.78 m. On RCP2.6, our projection of median SLR is 0.38 m in 2100, compared with 0.44 m in IPCC AR6. On RCP8.5, our projection of median SLR is 0.62 m in 2100, compared with 0.77 m in the IPCC AR6. This implies SLR from sources other than the AIS is lower in META than in IPCC AR6. Figures SI7 and SI8 provide PDFs/CDFs and survival functions, respectively, for all the projections. Table SI1 provides a hindcast test. Reference 42 provides probabilistic projections of SLR from Antarctica using the latest Shared Socioeconomic Pathways (SSPs).

## Coastal impacts

Figure 3 plots global economic costs of SLR from AIS melting, looking at two different response scenarios: no adaptation, and least-cost/optimal adaptation. As its name suggests, the former scenario involves no protection measures (e.g., physical defences) and no proactive retreat. Instead, SLR causes inundation, which in turn necessitates reactive retreat, and it raises the coastal floodplain, increasing the probability of storm damage. The latter scenario minimises total discounted costs of all types over a planning horizon of c. 40 years and chooses protection and/or proactive retreat where they help achieve this objective. Protection keeps the rising seas at bay through the construction of costly dikes, seawalls, etc. There remains some risk of overtopping during a storm surge. Retreat leaves land to be permanently inundated and also incurs relocation costs, while storm surges remain possible. Retreat is more likely to be chosen when the exposed land is of lower value. These scenarios can be regarded as upper and lower bounds on costs respectively: the reality is likely to be somewhere in between, due to chronic barriers to efficient adaptation[43,44]. Because the costs of SLR from AIS melting are unlikely to be independent of SLR from other sources, these estimates are obtained by differencing the results of running the CIAM model with all sources of SLR and running it with all sources of SLR except AIS melting.

Under the no adaptation scenario (panels a, c and e), global costs of incremental SLR from AIS melting are $180bn per year in 2050, rising steeply to $1.04trn/yr in 2100 on RCP4.5 (0.1% of global GDP). All results are reported in 2020 US dollars. On RCP2.6, the equivalent costs are $167bn/yr and $911bn/yr respectively, while on RCP8.5 they are $201bn/yr and $817bn/yr respectively. Figure 3 hence shows that AIS melting rapidly becomes a bigger economic problem towards the end of this century. Table SI4 shows that this is more a feature of AIS melting than SLR from other sources, at least for RCP2.6 and RCP4.5. Annual flow costs happen to be higher in 2100 on RCP4.5 than RCP8.5. This is a temporary phenomenon and just reflects the fact that, over short periods of time, costs of incremental SLR from AIS melting can

be non-monotonic. In turn, this is due to sequencing: faster SLR due to the extra contribution from the AIS can cause impacts to happen earlier. If we instead measure costs on a discounted net present value basis over a long period of time, RCP8.5 costs are higher than RCP4.5 costs. This is true both of the period up to 2100 and 2200 (see below, Fig. 5).

Permanent inundation, relocation/retreat, and flooding from storm surges contribute most—and broadly similar shares—to total costs in the no adaptation scenario, but the time profiles are different. Costs from permanent inundation and from relocation/retreat are incurred earlier and increase slowly over the century, however, flood/storm costs increase more quickly to become the dominant cost category towards the end of the century.

Optimal coastal protection/retreat has the potential to reduce costs hugely, as previous studies have emphasised[17,20]. Global costs of incremental SLR from AIS melting are roughly an order of magnitude smaller (panels b, d, and f). On RCP4.5 they are $23bn/yr in 2050, rising to $86bn/yr in 2100. On RCP2.6, the equivalent costs are $23bn/yr and $66bn/yr respectively, while on RCP8.5 they are $24bn/yr and $126bn/yr respectively. There is a particularly steep increase at the end of the century. Protection costs incurred in 2100 are in anticipation of impacts in the following century. Protection costs are generally small relative to retreat and residual damages, although they are not trivial and on RCP8.5 they reach $29bn/yr in 2100.

Comparing the error bars in Fig. 3 with the differences between panels, it is evident that there is much more cost uncertainty stemming from AIS uncertainty, conditional on the RCP scenario, than there is stemming from the RCP scenario itself. Moreover, cost uncertainty due to AIS uncertainty is large and increasing over time. Usually, the 90% C.I. is larger in absolute terms than the median cost estimate and it is particularly wide on RCP8.5 at the end of the century. Cost uncertainty conditional on a given amount of global mean SLR, due for example to uncertainty about engineering costs and behavioural responses, is not quantified here and will increase overall cost uncertainty.

Figure 4 shows the distribution of costs from AIS melting across countries in the middle of this century and makes clear that these are highly unequal, with a small number of countries experiencing large costs relative to GDP and the remainder experiencing low costs. The countries with high costs are almost all Small Island Developing States (SIDS). The exception among the 10 countries with the highest costs is the Netherlands, but only in the no adaptation scenario (panel a), which one might consider highly unrealistic in the Netherlands' case. These results are consistent with findings from CIAM for SLR from all sources[20], hence they are primarily explained by the extent of exposed coastline/assets.

In 2100, many of the countries experiencing the highest costs relative to GDP are again SIDS, but by this time the picture is more nuanced (see Fig. SI11). Assuming no adaptation, several countries in the Middle East and African region are in the top ten. Assuming least-cost adaptation, Australia faces notably high costs relative to GDP, as its assets/population are mostly located by the coast, and GDP grows more slowly than in lower-income countries, thus absolute costs matter more.

## Hydrofracturing, MICI, and a worst-case scenario

The estimates presented thus far exclude dynamic contributions of AIS to SLR triggered by hydrofracturing and MICI. More broadly, there is deep uncertainty about disintegration of marine ice shelves on Antarctica, and very rapid SLR from AIS melting is considered possible[29]. Recent research provides some indication that uncertainty about SLR from AIS melting might be underestimated in most of the ice sheet modelling literature[45], particularly the tail of high SLR outcomes, which is also the implication of the "low confidence" projection in IPCC AR6[29]. To take these concerns into account, here we complement our main estimates with two sensitivity analyses. The first uses the SLR

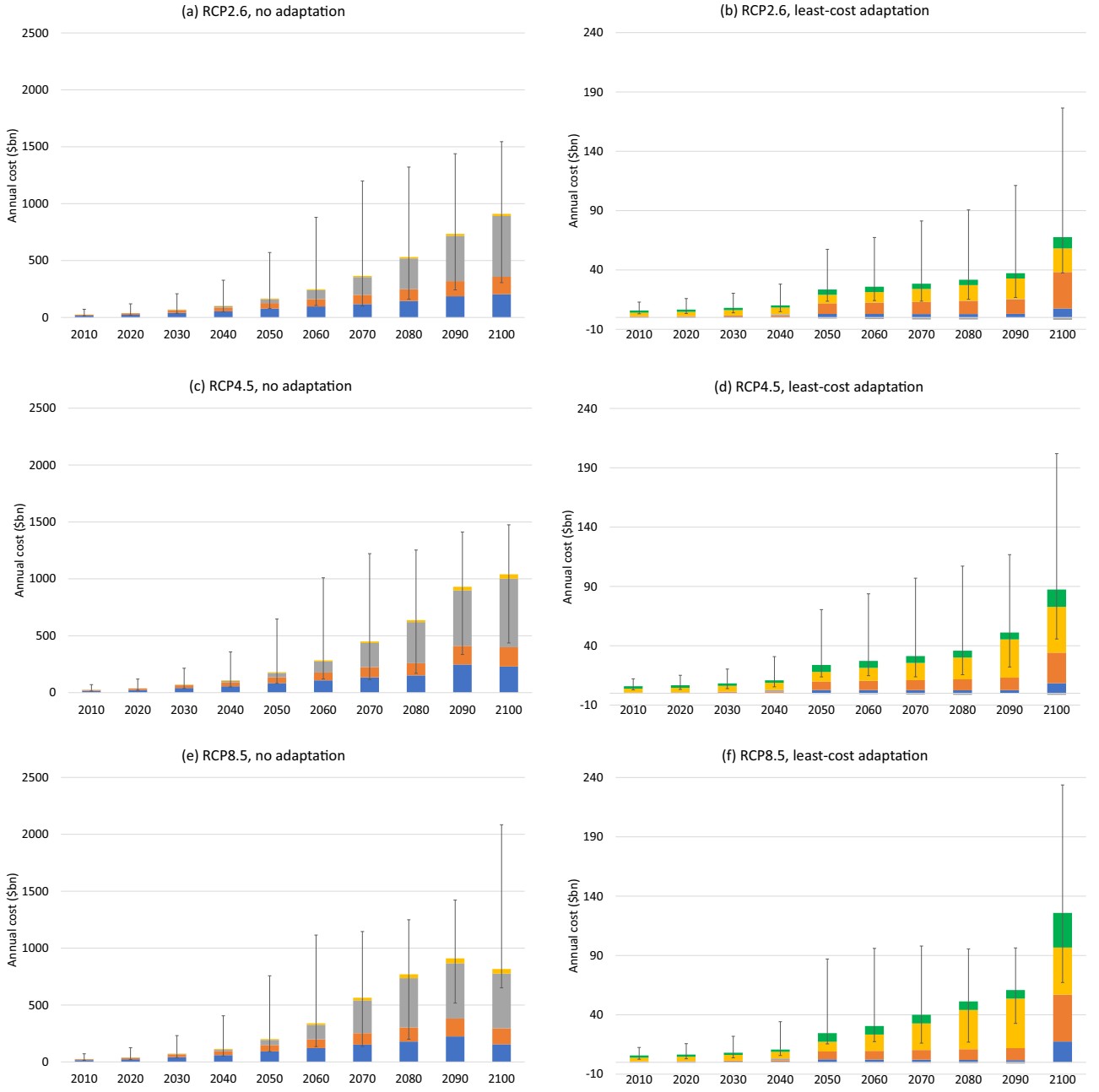

**Fig. 3 | Global annual adaptation and residual damage costs in 2020 US dollars, assuming no adaptation and least-cost adaptation, for different RCP scenarios. a** RCP2.6, no adaptation; **b** RCP2.6, least-cost adaptation; **c** RCP4.5, no adaptation; **d** RCP4.5, least-cost adaptation; **e** RCP8.5, no adaptation; **f** RCP8.5, least-cost adaptation. Error bars show the 90% confidence interval arising from global mean SLR uncertainty due to AIS melting.

projections of DeConto et al.[26], who incorporate hydrofracturing and MICI processes into a hybrid ice sheet-shelf model. The second uses the SLR projections of the ABUMIP project[27], which forced 15 ice sheet models with an extremely high melt rate underneath the ice shelves. The DeConto et al. projections are on RCP8.5. The ABUMIP experiment was not associated with any particular RCP, but again they are most plausibly associated with RCP8.5.

Figure 5 shows the discounted net present value (NPV) costs of SLR from these two sets of projections up to 2100, (4% consumption discount rate) and compares them with our main scenarios. Figure SI10 repeats the analysis on a 2200 horizon. The NPV costs of the DeConto et al. SLR projections are similar to our main estimates on a 2100 horizon. The NPV costs of the ABUMIP projections are multiple times higher, particularly on a 2100 horizon and under no adaptation, where the median estimate is

$18.8trn, or roughly 22% of world GDP in 2020. Figure SI9 shows the SLR projections underpinning these sensitivity analyses. The DeConto et al. projections are comparable to our main SLR projections, if somewhat higher in terms of the mean and median in 2100, but the C.I. is narrower, plausibly due to being based on only one ice sheet model. The ABUMIP projections are naturally much higher with much greater uncertainty (the 95% C.I. is literally off the charts). They are not considered realistic[27] but might be considered a worst-case scenario.

**Social cost of carbon**

Figure 6 shows the contribution of SLR from AIS melting to the social cost of carbon, estimated using the META IAM. The distribution of effects is obtained by sampling random parameters not just from the AIS melting model but also from the wider META model, quantifying

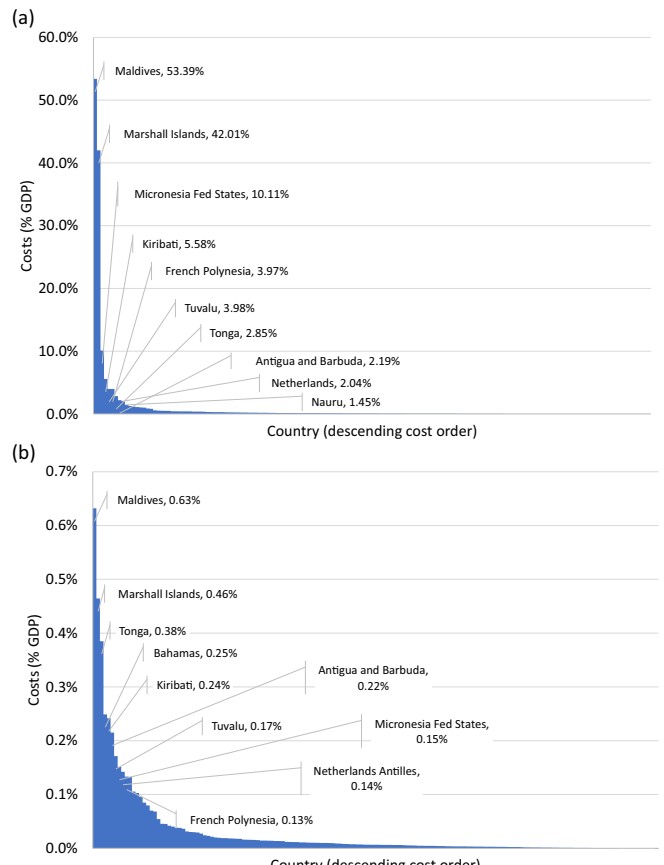

**Fig. 4 | Country-level costs of AIS melting as a percentage of GDP in mid-century on RCP4.5. a** no adaptation; **b** least-cost adaptation. Mid-century is defined as the 2040–2060 average.

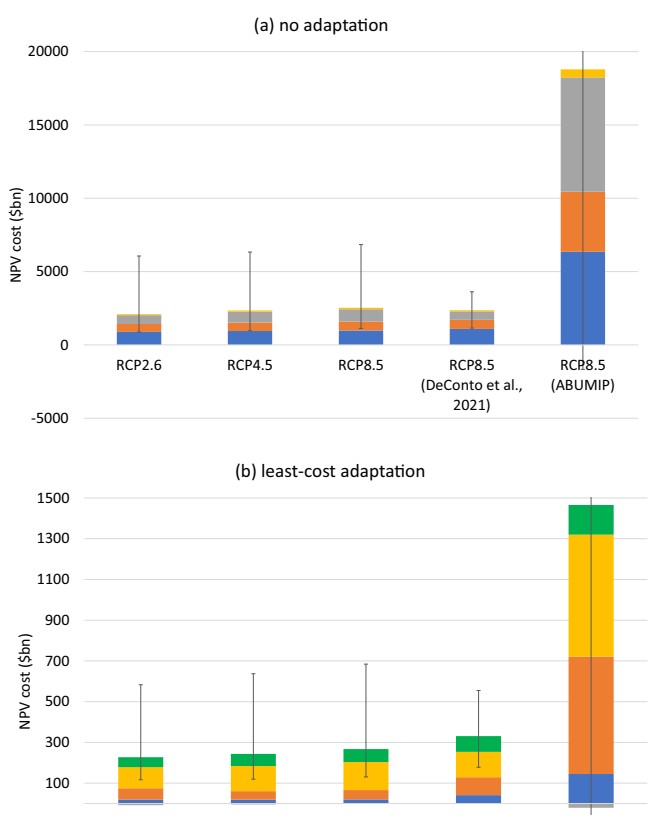

**Fig. 5 | Discounted net present value of global adaptation and residual damage costs up to 2100, assuming no adaptation and least-cost adaptation, for different scenarios. a** No adaptation; **b** least-cost adaptation. Error bars show the 90% confidence interval arising from global mean SLR uncertainty due to AIS melting. The 95% C.I. is truncated for the ABUMIP scenario. NPV costs are reported in 2020 US dollars using a 4% discount rate.

both climatic and economic uncertainties. We simplify CIAM to the level of national damage functions for incorporation in META. Each country in META has a single, national-level SLR damage function, which is linear with a slope coefficient that is calibrated on costs estimated by the full CIAM model at the segment level. Each coefficient is a random parameter that is symmetrically triangular distributed, with a maximum corresponding to country costs in the no adaptation scenario and a minimum corresponding to country costs in the least-cost adaptation scenario. Thus, the degree of adaptation is treated as uncertain and most probability mass is in between the extremes of the two scenarios presented in the analysis above. See the SI for further details.

The effect on the social cost of carbon of SLR from AIS melting is subject to substantial uncertainty but is positive in expectations, substantially so on high emissions scenarios. On RCP2.6-SSP1, the social cost of carbon increases by 7.1% on average, from a base of c. $34/tCO2. On RCP4.5-SSP2, it increases by 7.0% on average, from a much higher base social cost of carbon of c. $52/tCO2. To put these estimates in the context of previous studies, the 7.1% increase in the social cost of carbon due to AIS melting on RCP4.5-SSP2 is roughly double the 2.9% increase estimated by replicating Diaz and Keller's model of WAIS melting[13] in META[8], and is higher than the 1.8% increase estimated by replicating Nordhaus' GIS study[28] in META.

On RCP8.5-SSP5, the average increase in the social cost of carbon is 53.3%, from a base value of c. $33/tCO2. Note that the base social cost of carbon is lowest on RCP8.5-SSP2, despite this being the highest emissions scenario. The reason is that this emissions scenario should be paired with the SSP5 socio-economic scenario[46], which has high consumption per capita, thus the marginal value of climate damages is

lower. Nonetheless, the impacts of SLR from AIS melting can be extreme on RCP8.5. The distribution has a strong positive skew, which is evident in the figure from the position of the mean change relative to the median and the interquartile range. In some model runs, the social cost of carbon is more than doubled.

While many climatic and economic uncertainties contribute to these results, we fix the discount rate, with a pure rate of time preference of 1%. Figure SI12 repeats this analysis for pure rates of time preference of 0.1 and 2%. With a 0.1% pure rate of time preference, the contribution of SLR from AIS melting to the social cost of carbon is larger and the upper tail grows longer, while the opposite is true for a 2% pure rate of time preference. This is consistent with the global costs of AIS melting growing large only in the relatively long term (cf. Fig. 3).

## Discussion

In this paper, we have integrated recent modelling of AIS melting with a detailed model of the coastal impacts of SLR (CIAM). We have then situated this coupled model within the larger framework of the META IAM so that we can quantify the relative contribution of AIS melting to the social cost of carbon. This last exercise should be seen in the context of recent attempts to estimate the extra social cost of carbon coming from climate tipping points[8,28].

Our SLR projections from AIS melting are close to those of IPCC AR6. Our analysis of coastal adaptation costs and residual damages leads to the following conclusions and policy implications. First, the overall economic cost of AIS melting can be reduced dramatically by protecting and proactively retreating from coastlines where these pass

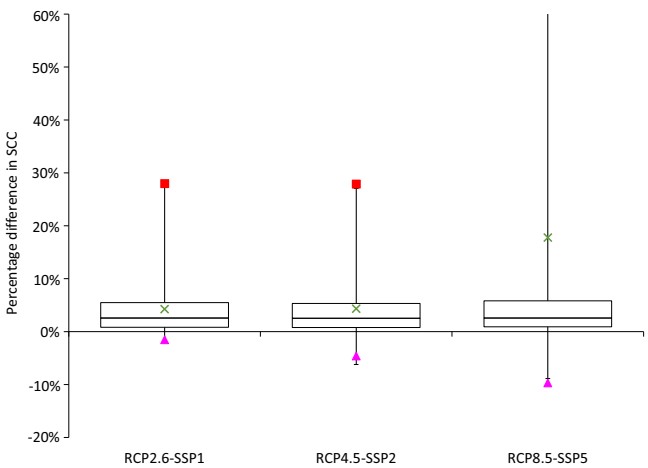

**Fig. 6 | Box and whisker plot showing the incremental contribution of SLR from AIS melting to the social cost of carbon.** Box shows median and interquartile range, whiskers show 95% CI, cross marks the average change (0.1% trimmed), triangle marks the 0.5 percentile, and square marks the 99.5 percentile. Standard META settings are used, i.e., pure rate of time preference of 1%, elasticity of marginal utility of 1.5, mixed levels/growth damages (phi = 0.5). Results from a Monte Carlo simulation with sample size 10,000.

a cost-benefit test. This result is consistent with previous studies looking at SLR from all sources[17,20]. Moving from a no adaptation scenario to a least-cost/optimal adaptation scenario reduces total costs by roughly an order of magnitude. Thus, our analysis underscores the importance of well-planned and well-resourced coastal adaptation and overcoming barriers to those. Second, the economic cost of AIS melting is extremely unevenly distributed worldwide. The costs are a very small proportion of the projected size of national economies in most countries, but they are considerable in around one to two dozen countries, primarily Small Island Developing States. Assuming no adaptation (or, by interpolation, imperfect adaptation), the costs look unmanageable in a few countries. This raises the issue of equity and what assistance these countries might receive to adapt to climate change and to cover residual loss and damage. Third, the cost of AIS melting is subject to large uncertainty stemming from the AIS contribution to SLR alone. This is evident in both the confidence intervals around our main estimates and our sensitivity analysis with an extreme scenario of ice shelf melting (ABUMIP). In turn, this leads us to speculate that there is a high economic value of reducing uncertainty through further research into AIS melting. Fourth, AIS melting typically adds several dollars to the social cost of every tonne of $CO_2$ emitted and there is a tail risk of very large increases in the social cost of carbon, particularly on the high emissions RCP8.5 scenario, where a doubling is feasible, and the expected increase is more than 50%. As anticipated by ref. 8, more comprehensive modelling of AIS melting leads to an upwards revision of the social cost of carbon and the contribution of tipping points to it.

Our research is subject to several limitations. Our analysis of AIS melting within META does not account for geophysical interactions with the GIS or other tipping elements such as the surface albedo feedback. Our quantification of SLR uncertainty is limited to the contribution from AIS melting (and, when run within META, to global mean SLR from all sources) and omits uncertainty from local sources. Moreover, our analysis omits uncertainty about the cost of SLR stemming from CIAM, which necessarily contains many uncertain assumptions and data points relating to the exposure of coastal economies and populations, and the costs of adaptation. For example, DIVA estimates of coastal flood exposure have been shown to be sensitive to changing the input data on extreme sea levels[47,48]. Overall, our confidence intervals are thus highly likely to be too narrow. In

addition, the cost of given uncertainty is not fully captured. For reasons of tractability, the local adaptation planner is assumed to have perfect foresight in CIAM. This means that our median cost estimates are based on planners optimally protecting coastlines and relocating based on the median SLR projection, similarly for the 5th and 95th percentiles. However, in reality planners may need to decide on coastal adaptation prior to the resolution of some of this uncertainty. Doing so typically requires hedging different outcomes and thus is costly. CIAM does not yet include all sources of cost from SLR (e.g., it omits saltwater intrusion), and it computes optimal adaptation strategies for each coastal segment independently. However, inter-dependencies may matter. See ref. 20 for reflections on how CIAM can be improved. META itself is situated within debates in climate economics about for instance discounting and the persistence of climate damages[8]. Future work could explore the economic implications of AIS hysteresis[24]. In principle, our dynamic ice loss model is capable of simulating hysteretic behaviour, but hysteresis hardly figures in the analysis of this paper because temperatures in the RCP scenarios are mostly monotonically increasing.

## Methods
### AIS melting model
We model SMB in reduced form. This firstly involves scaling global mean surface temperature, defined by the IPCC RCP scenarios, to continental-scale temperature on Antarctica using the results of ref. 24. Secondly, continental-scale warming is related to changes in accumulation using the analysis of ref. 23, which is based on ice-core data, paleo-simulations, future simulations from the CMIP5 general circulation models, and from one high-resolution, regional climate model. This analysis suggests an increase in accumulation of 5+/−1% per degree continental warming. Thirdly, again using[23], accumulation is related to total AIS mass balance taking into account a lagged interaction with dynamic processes. Lastly, an adjustment factor is applied to represent evidence that the SMB of the ice sheet will turn negative at approx. 6.5 K warming above pre-industrial[24]. See the SI for further details.

We model dynamic contributions using the reduced-form model of ref. 25, which is designed to emulate basal ice shelf melting and the resulting contribution of the AIS to SLR in 16 state-of-the-art ice sheet models (the LARMIP-2 models). This firstly involves scaling global mean surface temperature to subsurface Antarctic ocean temperatures using CMIP5 data. Secondly, these subsurface ocean temperatures are mapped into basal ice shelf melting using observational data. Thirdly, basal ice shelf melting is mapped into SLR using response functions emulating the behaviour of each of the 16 ice sheet models. The analysis is conducted separately for the five major ice basins on the continent: East Antarctica, the Ross Sea, the Amundsen Sea, the Weddell Sea, and the Antarctic Peninsula. The total contribution to SLR from AIS dynamic processes is the sum of the regions' contributions. See the SI for further details.

Although basal ice shelf melting is considered the most important dynamic process on Antarctica in the near term[25], there are other dynamic processes that are omitted by our model, notably ice shelf hydrofracturing and Marine Ice Cliff Instability (MICI)[26,35–37]. Reference 36 projected very limited hydrofracture potential this century outside the Antarctic Peninsula, even in high warming scenarios. In a recent model intercomparison project ref. 37 found that the addition of ice shelf collapse based on hydrofracture had very little effect on SLR from Antarctica. However, ref. 26, have documented rapid rates of mass loss in quaternary records that have not previously been replicated by models without invoking MICI and ice hydrofracture processes, or by changing other aspects of the modelling such as ice sliding laws. They suggest that MICI and hydrofracture processes are the most geophysically tractable mechanisms for ice sheet models to replicate rapid rates of ice loss from Antarctica, such as during the

Last Interglacial. Therefore, to supplement our results, we also run the published SLR projections of ref. 26 through our modelling steps below. These results are derived from an ice sheet model that includes ice shelf hydrofracturing and MICI, and they are available for the high emissions RCP8.5 scenario.

Other forcings that may affect AIS dynamics include basal lubrication of outlet glaciers enabling sliding over the bed, ice softening that enhances movement through ice creep, and surface elevation changes through altered precipitation, in turn changing the hypsometry of the glacier.

### Statistical downscaling of global SLR to local SLR

The coastal impacts model we use (ref. 20; *see below*) builds on the Dynamic Interactive Vulnerability Assessment (DIVA) database[38], which partitions the world's coastlines into 12,148 segments with homogeneous physical characteristics. The median length of a segment is 18 km. Therefore, we must downscale our global SLR projections to the segment or local level. We do this by statistical downscaling. That is, we estimate local SLR functions that we fit to the relationship between local and global mean SLR according to the local SLR projections of ref. 22. In each of the 12,148 segments, we specify the absolute difference between local and global SLR as a flexible, cubic function of global SLR and minimise the squared difference between the function and the data from[22]. Further details of the procedure can be found in the SI, where we also evaluate its performance in terms of goodness of fit. On the middle-of-the-road RCP4.5 scenario, the median absolute error of the local SLR functions is 0.003 m computed over the period 2010–2100 and over all segments (mean absolute error = 0.005 m; 97.5% of predictions within 0.018 m), thus the fit is good.

### Coastal adaptation and damage costs

As mentioned, we estimate coastal adaptation and damage costs using the Coastal Impact and Adaptation Model (CIAM) of ref. 20. At the segment level, a CIAM planner determines whether to adapt to SLR by minimising the sum of discounted adaptation costs and residual damages over a rolling planning horizon (standardly 40 years). Adaptation costs come in two forms: protection by e.g., dikes and sea walls, or managed retreat through relocation of people and infrastructure. Residual damages come in three forms: the value of land lost to permanent inundation; the value of lost wetland ecosystems, valued at willingness to pay for wetland ecosystem services; and flood damage from storm surges. The optimal (least-cost) strategy for each segment is based on local physical and socio-economic characteristics, and the local SLR projections.

### Wider IAM framework

The model elements described so far take global mean surface temperature scenarios as their exogenous input and eventually calculate the economic costs of SLR from AIS melting at the level of small segments of coastline, which can then be aggregated to the national and global levels. We then take this model of AIS impacts and place it, as a module, within the larger META (Model of Economic Tipping point Analysis) Integrated Assessment Model of ref. 8, version 2021.

META is a modular IAM, designed to facilitate studying the economic impacts of tipping points. We take this approach for two reasons. Firstly, we want to account for the possible dependence of the impacts of AIS melting on the SLR contribution from other sources, namely thermal expansion of the oceans, melting of glaciers and small ice caps, and GIS melting. That is, we do not want to assume AIS impacts are additively separable. Because global mean surface temperature is the primary driver of SLR from other sources, it is not likely that SLR from one source has a major effect on SLR from another. Rather, it is likely to arise on the cost side, through, e.g., the effect of coastal topography. META allows this dependence because it contains

separate modules representing these other sources of SLR. The contribution from thermal expansion, glaciers and small ice caps is based on the projected range published in IPCC AR5[49]. The contribution from GIS melting is based on replicating the model of ref. 28. Thus, the economic impacts of AIS melting are ultimately estimated not in isolation, but rather as the difference between impacts from all sources of SLR and impacts from all sources of SLR less AIS melting. Secondly, the wider META framework includes projections of global mean surface temperature, economic and population growth (from the RCP/SSP scenario set), and enables the impact of temperature and SLR damages on national consumption per capita to be estimated. This then allows us to estimate the contribution of AIS melting to the social cost of carbon, that is, the present value of the economic costs of emitting an extra tonne of carbon dioxide, a key input to policy processes worldwide[39,40].

### Estimating the social cost of carbon

The incremental contribution of AIS melting to the social cost of carbon —the social cost of AIS melting—is calculated by running the META IAM with and without a contribution to SLR from AIS melting, estimating the social cost of carbon in both cases, and then taking the difference. The social cost of carbon is the present value of the economic costs of emitting an extra tonne of carbon dioxide. It is formally the monetary equivalent of the loss in global social welfare from the extra tonne of $CO_2$[50]. Global social welfare is standardly calculated as the discounted sum of national population times national utility per capita. National utility per capita is calculated by applying a utility function to national consumption per capita, net of the costs of climate change. See ref. 8 for a full description of the welfare calculations in META.

Estimating the social cost of carbon requires a simplified approach to SLR costs relative to the full CIAM model; this approach is set out in the SI. In essence, an SLR damage function is specified for each country, with a probabilistic cost parameter calibrated on the full CIAM model under no protection as an upper bound and optimal/least-cost adaptation as a lower bound (thus, in most contingencies protection lies in between). The social cost of carbon is calculated for an emissions impulse in 2020 and is given in 2020 US dollars.

### Treatment of uncertainty

Our probabilistic estimates of SLR are generated by coupled Monte Carlo simulation of the AIS SMB and dynamic models, together with SLR from other sources where total SLR is being estimated. Uncertainty comes from random sampling of probabilistic parameters as described in the SI. We take a sample of 50,000. Table SI3 shows that this is sufficient for numerical convergence.

The 5th, 50th, and 95th percentiles of the resulting distributions of SLR are then inputted to CIAM and optimal adaptation choices (protection and retreat in the case of the overall optimal/least-cost scenario, retreat only in the case of the no protection scenario) are computed conditional on the percentile in question.

META includes simplified national-level SLR damage functions. When computing the social cost of carbon, the full META model including the AIS module is subject to a Monte Carlo simulation. Uncertainty comes from the random parameters of the AIS module as well as many other random parameters as described in ref. 8.

## Data availability

The data generated in this study have been deposited in the Zenodo database under accession code https://doi.org/10.5281/zenodo.7075824.

## Code availability

Model code and documentation for CIAM are available at https://github.com/delavane/CIAM and for META at https://github.com/openmodels/META-2021.

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

## Acknowledgements

We are indebted to Julius Garbe and Anders Levermann for helping us apply their work. James Rising, Thomas Stoerk and Gernot Wagner co-developed the META IAM with S.D. We are grateful to Klaus Keller, Bill Nordhaus, James Rising, Glenn Rudebusch, and Thomas Stoerk for comments. S.D. gratefully acknowledges the financial support of the Grantham Foundation for the Protection of the Environment and the ESRC Centre for Climate Change Economics and Policy. F.K. contributed to this work while working at the Grantham Research Institute on Climate Change and the Environment and prior to joining Arup. This article should not be reported as representing the official views of Arup or its members. The opinions expressed and arguments employed are those of the authors.

## Author contributions

F.K. and S.D. conceived of the paper. F.K. developed the AIS model. S.D. coupled this to the CIAM and META models. F.K. and S.D. wrote the paper.

## Competing interests

The authors declare no competing interests.
