## [Peer review file · Nature Communications]

REVIEWER COMMENTS

Reviewer #1 (Remarks to the Author):

The manuscript is overall reasonably well written and contains sufficient detail to perform this review. The manuscript seems to have the potential to rise to the level of quality one expects in this journal. However, there are several loose ends that need to be addressed before this manuscript is ready to be accepted. Below, I enumerate key issues that need to be addressed.

Specific points:

1. The abstract is long on points that are arguably already well made in the literature. The last four to five lines focus on the new contributions. It would be helpful to focus the abstract more on what is actually new.
2. line 55: To claim that these studies are disconnected without a qualifier seems to be an overstatement.
3. Figure 1: It would be extremely helpful to have a representation of what are the key uncertainties that are considered in this workflow.
4. Page 5: Please show the PDF, the CDF, and the survival function of the projections.
5. Please show a pairs plot of all parameters that have been considered as uncertain.
6. Please include a hindcast test to allow the reader to assess how well the model succeeds in approximating the observations.
7. Please provide evidence that the sample size of 50,000 is sufficient for numerical convergence.
8. Line 193: Please express this also as percent of gross world product. Further, please explain whether gross world product is considered to change and - if so - by which scenario.
9. Figure 3: Please make the color separation between protection and retreat more clear.
10. Figure 4:
 - a. The text suggests that this is from the AIS melting alone, but the caption is silent on this. Please clarify this in the caption.
 - b. Please specify whether GDP is dynamic or static. If it is dynamic, what are the assumptions?
 - c. There is a lot of white space in this figure. Can the figure be improved?

- d. For the lower panel, please provide some detail how the adaptation is actually occurring (in the model) in cases such as the Maldives, the Marshall Islands, and Tuvalu.
11. Overall, many figure designs need work to abide by the rather specific format prescription of this journal.
 12. line 287: Please provide the required details for the uncertainty analysis.
 13. Do we really believe four significant digits for an estimated social cost of carbon?
 14. Line 296: What are hypotheses to potentially explain the differences from the discussed previous studies?
 15. Line 321: What are these certain economic studies?
 16. Line 329: There are many more academic fields involved than just glaciology.
 17. line 354: Please link back to evidence provided in this study for this statement.
 18. Please provide information on code and data availability. This includes the repository URL, the license, access to potentially required data, and code to replicate the study.
 19. Supplementary material figure 3: This figure (and others in the SOM) should be labeled with an indicator that they are in the SOM. These are, I presume, results from the emulator. If this is correct, please provide a comparison to the data used to fit this emulator.

Reviewer #2 (Remarks to the Author):

The manuscript assesses the economic cost of sea level rise from the ice loss of the Antarctic ice sheet. The authors introduce parametrizations for Antarctic dynamic ice loss and surface mass balance dependent on global mean temperature change that are rooted in recent advances in process-based modelling. They assess economic damages by deriving national damage functions from more detailed data from the CIAM model. Results suggest importance of Antarctic ice loss for the social cost of carbon and that overall damages can be large for a number of vulnerable countries. The dependence on the choice of adaptation or no adaptation is extreme.

The topic of the manuscript is important and timely. There are only few studies that globally assess the cost of sea level rise or the Antarctic contribution to such cost. The approach to use a simple climate model and parametrizations for Antarctic ice loss derived from more detailed modelling is well presented. The uncertainty of future Antarctic ice loss is well addressed. However, how the authors approach the translation of the further causal chain from sea level rise to damages is a major weakness of the study.

First, they rely heavily on a single framework, CIAM. While the manuscript spends much detail on the difficulties to capture the uncertainties in the physical part of the emissions-to-damages chain, the assumption of CIAM being close to truth seems taken for granted and is not discussed. CIAM builds on the DIVA model and it has known deficiencies, see for example Muis et al 2017 (i.e. section 3.4.2) and Woodworth et al. (2017).

Second, it is not clear how the damage functions are derived and how valid can the interpolation be between the highly idealistic perfect adaptation and no adaptation scenarios of CIAM/DIVA. The aggregation to national damages seems valid, but how can you ensure that your interpolated results between the adaptation / no adaptation extremes can be trusted? There is only some explanation in the SI (less than 15 lines), but no assessment and discussion of what uncertainty enters through the approach and how trustworthy it is. Minor: you also need to dig into the only methods and then SI to understand that for the calculation of the social cost of carbon an interpolation is used. How do you sample from the continuum between adaptation and no adaptation?

It is the adaptation/no adaptation choice, which adds most uncertainty to the damage estimates, spanning two orders of magnitude (in Fig. 4). This dwarfs the uncertainty from unknown response of the ice sheet and emission scenarios, which questions the balance of the manuscript: why spend so much detail on ice sheet and emission scenario uncertainty when most of it is from the adaptation/no adaptation.

The study thus critically needs to address the above aspects, I otherwise do not find the damage estimates and the conclusions drawn from them worth presenting in Nature Comms.

A second, but less critical point is the necessity of the presented sea level model within the study. The study sticks to the standard RCP scenarios for which IPCC sea level numbers are available. SLR emulators like the one presented are most useful if you need to assess different scenarios than the standard ones. The economic assessment as such could be done with the official IPCC AR6 sea level estimates directly. It would be more straightforward and better for comparability to first use the study design with the standard IPCC numbers and maybe then explore additionally extreme scenarios like ABUMIP.

Minor comments:

title: "melting suggests the wrong process: it is mainly ice loss to the ocean"

I24: "several dollars": needs relation to the total estimated cost of carbon

I26: more than what? total cost of carbon?

I41: unclear: storm surge or permanent inundation are not residual damages

I83: mass balance change?

I265: what are NPV costs?

I312: now 1% discount rate, not 4%?

I338: conclusion also true already from DIVA 2014

I452-I458: exactly: impacts are not easily separable per SLR contribution, but you do this for AIS in your main analysis.

I463: your numbers thus depend also on all other components except the system is linear. therefore the numbers for total SLR should be given.

I468: impact on consumption per capita? this is a first time mention. before, the reader thinks it is all about damages and adaptation direct costs.

I S205: Country-level damages from SLR: good idea, but this completely relies on CIAM. There is a two order magnitude of adaptation vs no adaptation. is interpolation credible?

References

Muis et al. 2017

<https://agupubs.onlinelibrary.wiley.com/doi/full/10.1002/2016EF000430>

Woodworth et al. 2017

<https://www.sciencedirect.com/science/article/abs/pii/S0921818116304933>

Reviewer #3 (Remarks to the Author):

The manuscript assesses the contribution of the melting of the Antarctic Ice Sheet (AIS) with a set of models tailored for the analysis in an integrated assessment framework. The authors build two reduced-forms models to describe the AIS behavior using data and information from recent literature. These two models are then coupled with the CIAM module integrated in the META (IAM) model to provide the economic cost of the AIS melting by comparing impacts from all sources of SLR with impacts from all sources of SLR less AIS melting.

Although the subject has already been studied in the past, as acknowledged in the manuscript, the differences highlighted by the authors are a contribution of new knowledge to the economic literature. Results show the importance of AIS in economic costs growing by the end of the century and that the AIS melting could increase the social cost of carbon from 7% to 53.3% on average depending on the scenario considered.

While there are some of points I think could be clarified and discussed in more detail in the manuscript, I recommend its publication after they have been addressed by the authors.

1. The fact that annual flow costs in 2100 are higher on RCP4.5 than on RCP8.5. Although the authors indicate that sequencing would be one of the main reasons explaining that result, it is not clear enough why costs in RCP 8.5 are decreasing after 2090, despite SLR is increasing even more. Could it be the negative contribution of SMB to SLR as shown in figure 2 or is it due to a reduction of population/asset exposure to SLR? Are there other reasons explaining that behavior?

2. It is not clear if the results shown for each RCP on the manuscript (figures 3-5) are in combination with a specific SSP or other socioeconomic data. Only in the SCC section there are RCP/SSP combinations. Please clarify this in the modelling approach section.

3. Line 297 mentions the combination RCP8.5-SSP2, but then lines 299-300 state that RCP8.5 should be paired with SSP5. I would encourage the authors to explain the choice of RCP8.5 with SSP2 and the implications for the SCC of an RCP8.5-SSP5 combination.

4. Finally, the following reference could be included in the introduction:

Edwards, T.L., Nowicki, S., Marzeion, B. et al. Projected land ice contributions to twenty-first-century sea level rise. *Nature* 593, 74–82 (2021). <https://doi.org/10.1038/s41586-021-03302-y>

Economic impacts of melting of the Antarctic Ice Sheet

Response to Reviewers

14th June 2022

We would like to sincerely thank the reviewers for their detailed and helpful comments on the manuscript. We have produced a thorough revision that, we hope, addresses all the points raised. Below we provide a point-by-point response to the reviewers' comments.

n.b. original comments in *italics*

Reviewer #1

The manuscript is overall reasonably well written and contains sufficient detail to perform this review. The manuscript seems to have the potential to rise to the level of quality one expects in this journal. However, there are several loose ends that need to be addressed before this manuscript is ready to be accepted. Below, I enumerate key issues that need to be addressed.

Thank you very much for your careful review of our paper. We hope we have tied up all the loose ends!

Specific points:

1. The abstract is long on points that are arguably already well made in the literature. The last four to five lines focus on the new contributions. It would be helpful to focus the abstract more on what is actually new.

This is fair. The first sentence makes some basic points about melting of the Antarctic Ice Sheet and its implications, so we assume this is the problem sentence (the next three sentences summarise what we do in the paper, and the remaining sentences summarise our results). The reason for sentence #1 is that we are trying to conform with the guidelines for how to draft a summary paragraph for a Nature journal. According to these, the abstract/summary paragraph should begin with "One or two sentences providing a basic introduction to the field, comprehensible to a scientist in any discipline." (http://www.cbs.umn.edu/sites/default/files/public/downloads/Annotated_Nature_abstract.pdf)

Thus, before we explain what we did, we think we need to set the scene.

2. line 55: To claim that these studies are disconnected without a qualifier seems to be an overstatement.

We agree and apologise for that. We have added a qualifier to explain what we mean: "However, these studies are disconnected from the actual process of ice sheet melting and SLR, because they work with abstract damage functions depending on global mean temperature."

3. Figure 1: It would be extremely helpful to have a representation of what are the key uncertainties that are considered in this workflow.

Thank you for this good suggestion, which we had not considered. We have added the key uncertainties that we model to the flowchart. They are mapped on to the model components, to the right of the workflow, in yellow.

4. Page 5: Please show the PDF, the CDF, and the survival function of the projections.

Thank you for the suggestion, we have added PDFs and CDFs to the Supplementary Information, Figure 7. We considered also providing survival functions as you have requested, but since our

model is not based on survival analysis, we concluded survival functions are poorly suited for quantifying the uncertainty.

5. *Please show a pairs plot of all parameters that have been considered as uncertain.*

Thank you for the suggestion, we have included these in the Supplementary Information as Figures 2-5. This exercise is complicated by the response functions in SI Equation (7), which are a key uncertainty. Recall that these emulate the dynamic behaviour of various complex ISMs and are taken from Levermann et al. (2020). Importantly, each is a time series/vector of ice sheet responses, rather than a scalar. Thus, there are in total 950 region/year values of the uncertain ice sheet response function, to add to a further 12 uncertain scalar parameters in the AIS melting model (comprising 4 parameters, 2 of which have a value in each of the 5 Antarctic regions). We reduce the dimensionality of this exercise by collapsing the time dimension in the response functions. We do this by computing the *cumulative* response in 2100 to deterministic basal ice shelf melting on the RCP4.5 scenario. That is, we fix all the other uncertain parameters, including in the warming module, not just the AIS melting model, to their central estimates and then compute the uncertainty in the cumulative ice sheet response.

6. *Please include a hindcast test to allow the reader to assess how well the model succeeds in approximating the observations.*

Thank you for this suggestion. Our modular framework integrates several models and only one of these can feasibly be hindcasted. This is the model of AIS dynamic contributions, which is by far the largest source of SLR from AIS melting in our framework. Fortunately, this has already been subject to a hindcast test in Levermann et al. (2020), Section 3. We now refer to this in the SI, discussing the results of the test on lines 84-87. Regarding the other model components, the quantitatively unimportant SMB contribution cannot be hindcasted because “[o]bservations over the past decades...are unable to constrain the relation between temperature and accumulation changes because both are dominated by strong natural variability” (Frieler et al., 2016, abstract). Similarly, there are no systematic observational data on coastal impacts of SLR with which the CIAM model could be hindcasted.

7. *Please provide evidence that the sample size of 50,000 is sufficient for numerical convergence.*

Thank you for this suggestion. SI Table 2 provides a new analysis of the consistency of our 2100 projections of SLR from Antarctica over 5 samples of size 50,000. For this analysis, we use RCP8.5 emissions, as this will provide the stiffest test of numerical convergence. Results agree to within at least 0.01m, depending on the percentile, and usually more closely than that.

8. *Line 193: Please express this also as percent of gross world product. Further, please explain whether gross world product is considered to change and - if so - by which scenario.*

We have done so -- \$1.04trn/yr in 2100 is about 0.1% of projected world GDP in that year. We considered whether to always quantify both the absolute monetary value and the percentage of world GDP, but we concluded this would add too much clutter and one mention is sufficient to give a sense of how large the costs are in relative terms.

World GDP changes over time, as it should. We clarify in the SI that in the CIAM model income growth is based on IMF projections and population growth on the UN projections.

9. *Figure 3: Please make the color separation between protection and retreat more clear.*

We have done this by changing the colour of protection to green.

10. *Figure 4:*

a. The text suggests that this is from the AIS melting alone, but the caption is silent on this. Please clarify this in the caption.

This is correct. We have clarified it.

b. Please specify whether GDP is dynamic or static. If it is dynamic, what are the assumptions?

GDP is dynamic in the sense that it grows over time (please refer to our response to point 8, above).

c. There is a lot of white space in this figure. Can the figure be improved?

You are right, there is a lot of white space. We have experimented with several different ways of visualising the country-level information and this seems like the best option to us. We investigated maps, but of course the major problem with that is that many of the countries with high costs are tiny in terms of area, thus tiny on a map. Nor does a map convey the skew on the distribution of costs across countries, which is a key result in our view. We also looked at a scatter plot in decreasing cost order, but a column chart is more visually effective in our view. This does leave white space, admittedly. We also experimented with a box where the white space now is. This box conveyed the same information that is now in the data labels. We concluded the labels did a better job of identifying the data points (columns). In sum, we don't think we can improve on the figures. Hopefully you see our working!

d. For the lower panel, please provide some detail how the adaptation is actually occurring (in the model) in cases such as the Maldives, the Marshall Islands, and Tuvalu.

There are two adaptation mechanisms in CIAM, protection and relocation/retreat to higher ground (which can be proactive or reactive; in the least-cost adaptation scenario it will usually be proactive). What combination of these measures ends up getting deployed is country- and time-specific, which makes it difficult to provide more details in a reasonably compact manner. The time-specificity is further complicated by the fact that planned adaptation mostly happens in advance, so adaptation costs are incurred in earlier periods to avoid residual damage costs in later periods. For these reasons, our preference is not to provide more details in relation to Figure 4. It is one of the main purposes of Figure 3 to show how costs are broken down and to illustrate both the role of adaptation and how it is done.

11. *Overall, many figure designs need work to abide by the rather specific format prescription of this journal.*

We apologise if this is so. May we suggest that we work with the editor to solve any problems? It may stem from our unfamiliarity with the journal, although we have published in Nature journals before without too many difficulties related to figures, and we have read the guidelines again and believe that we comply with them.

12. *line 287: Please provide the required details for the uncertainty analysis.*

The uncertainty analysis of the AIS melting model is now fully described in the Supplementary Information, and the details of how the uncertainty analysis is undertaken for the wider META model is fully described in Dietz et al. (2021). The subsection of the Methods section entitled "Treatment of uncertainty" summarises where the reader can find the different pieces of information.

13. *Do we really believe four significant digits for an estimated social cost of carbon?*

This is a fair point. We have rounded up to the nearest dollar.

14. Line 296: *What are hypotheses to potentially explain the differences from the discussed previous studies?*

This is a great question, thank you for pushing us to explain the differences! The explanation is actually pretty simple. One of the advantages of the META framework in this regard is that it controls for all other relevant differences. That is, you can keep all the other modules in the IAM constant and just change the ice sheet module(s). This means any differences are entirely explained by shifts in the projected SLR distribution. We have added a sentence on this.

15. Line 321: *What are these certain economic studies?*

We have now deleted this part of the discussion in the interests of brevity.

16. Line 329: *There are many more academic fields involved than just glaciology.*

We have now deleted this part of the discussion in the interests of brevity.

17. line 354: *Please link back to evidence provided in this study for this statement.*

You are right to pick us up on this. We provide no direct evidence that there is “a high economic value of reducing uncertainty through further research into AIS melting”. However, since the theory of the value of information, going back to Blackwell’s work in the 1950s, has is that a reduction in uncertainty is valuable in most circumstances, it seems like an obvious implication of our findings of large uncertainty. We have modified the language to make clear that here we are speculating, which we trust is not unreasonably since this is the discussion section. We now write “this leads us to speculate that there is a high economic value of reducing uncertainty through further research into AIS melting”.

18. *Please provide information on code and data availability. This includes the repository URL, the license, access to potentially required data, and code to replicate the study.*

Certainly, we have added a code availability section before the references, per the journal’s guidelines. Model code and documentation for CIAM are available at <https://github.com/delavane/CIAM> and for META at <https://github.com/openmodels/META-2021>.

We will also make all the data underpinning the figures in the manuscript publicly available before publication (url to be confirmed), if we manage to progress to the next stage. We have added a data availability section before the references, again per the journal’s guidelines, with a placeholder for the url.

19. *Supplementary material figure 3: This figure (and others in the SOM) should be labeled with an indicator that they are in the SOM. These are, I presume, results from the emulator. If this is correct, please provide a comparison to the data used to fit this emulator.*

We have re-labelled all the SI figures and tables.

No, these are not results from the emulator, in fact. For the DeConto et al. SLR projections, we simply replicate what they did. That is, we obtained the ‘raw’ data underpinning their published SLR distribution, namely SLR projections for each of 109 different parameterisations of their ISM. We then replicated their procedure of assigning equal probability to each of the 109 parameterisations and running a large N Monte Carlo simulation. So, the distribution we end up with is identical to DeConto et al. (to within a very small sampling error). The Sun et al. ABUMIP study does not provide a probability distribution *per se* for future SLR, rather they report the projections of each participating model, and we construct a probability distribution over the models by adding our own assumption that each model has equal probability and again running a large N Monte Carlo

simulation. Consequently, the range of our SLR distribution corresponds exactly to the difference between the lowest and highest models in Sun et al. figure 1(b), with the median towards the lower end of the range since most models are clustered there. We have added this comparison to the text just before the figure.

Reviewer #2

The manuscript assesses the economic cost of sea level rise from the ice loss of the Antarctic ice sheet. The authors introduce parametrizations for Antarctic dynamic ice loss and surface mass balance dependent on global mean temperature change that are rooted in recent advances in process-based modelling. They assess economic damages by deriving national damage functions from more detailed data from the CIAM model. Results suggest importance of Antarctic ice loss for the social cost of carbon and that overall damages can be large for a number of vulnerable countries. The dependence on the choice of adaptation or no adaptation is extreme.

The topic of the manuscript is important and timely. There are only few studies that globally assess the cost of sea level rise or the Antarctic contribution to such cost. The approach to use a simple climate model and parametrizations for Antarctic ice loss derived from more detailed modelling is well presented. The uncertainty of future Antarctic ice loss is well addressed. However, how the authors approach the translation of the further causal chain from sea level rise to damages is a major weakness of the study.

First, they rely heavily on a single framework, CIAM. While the manuscript spends much detail on the difficulties to capture the uncertainties in the physical part of the emissions-to-damages chain, the assumption of CIAM being close to truth seems taken for granted and is not discussed. CIAM builds on the DIVA model and it has known deficiencies, see for example Muis et al 2017 (i.e. section 3.4.2) and Woodworth et al. (2017).

You are quite right to point this out. It was a major oversight on our part not to draw the reader's attention to the additional uncertainties contributed by the CIAM/DIVA model. Because CIAM/DIVA is a rich modelling framework, the uncertainties are numerous! We have added a new passage to the discussion section, which is where we list the main limitations of our study (line 350 onwards). Thank you for the reading suggestions: we read these papers and cite them.

There is obviously much more that one could say, but we are under severe space constraints. Hopefully you will agree that this suffices. Our reliance on CIAM/DIVA is certainly a weakness, but relative to the ice sheet modelling literature there is a lack of alternatives that could easily be run alongside CIAM/DIVA in a comparison. In general, our observation as relatively new entrants to this literature on economic impacts of SLR is that it would benefit from a greater number of models, updated more frequently. It has not attracted as many scholars or resources as it needs and as it should.

Lastly, in the spirit of not overselling our results and using CIAM uncritically, we have deleted the following passage from the discussion section: "The fundamental aim of this paper has been to estimate the economic impacts of melting of the AIS, via SLR, integrating the modelling state-of-the-art in glaciology and climate economics (emphasis added)".

Second, it is not clear how the damage functions are derived and how valid can the interpolation be between the highly idealistic perfect adaptation and no adaptation scenarios of CIAM/DIVA. The aggregation to national damages seems valid, but how can you ensure that your interpolated results between the adaptation / no adaptation extremes can be trusted? There is only some explanation in the SI (less than 15 lines), but no assessment and discussion of what uncertainty enters through the approach and how trustworthy it is. Minor: you also need to dig into the only methods and then SI to understand that for the calculation of the social cost

of carbon an interpolation is used. How do you sample from the continuum between adaptation and no adaptation?

It is the adaptation/no adaptation choice, which adds most uncertainty to the damage estimates, spanning two orders of magnitude (in Fig. 4). This dwarfs the uncertainty from unknown response of the ice sheet and emission scenarios, which questions the balance of the manuscript: why spend so much detail on ice sheet and emission scenario uncertainty when most of it is from the adaptation/no adaptation.

First, we apologise for the need to dig deep into the SI for this information, but the format of the journal precludes us from engaging with a detailed discussion of methods except in the SI, and to a lesser extent in the methods section of the main paper (which is still heavily restricted). We try to remedy this somewhat by adding a sentence to the main paper on p4, which states that for the calculation of the social cost of carbon we “treat the degree of adaptation as an uncertain parameter at the country level.”

As for our approach, we do feel that it makes sense. Logically, the no adaptation and least-cost adaptation scenarios in CIAM constitute upper and lower bounds respectively on the overall costs of SLR. It cannot be possible to do better, economically, than choosing protection and proactive retreat so as to minimise overall costs. And it cannot be possible to do worse than never choosing protection or proactive retreat (the implicit assumption in a planning framework like this is that adaptation, when undertaken, is effective at reducing costs and does not result in costs higher than they would have been absent adaptation). We then assume that in all likelihood countries will do something in between, i.e., they will adapt, but sub-optimally due to well-documented barriers to adaptation that we cite in the main paper. There are essentially no data that could guide the specification of the distribution of the country cost coefficients that embody adaptation, so we look to the most familiar specifications that one would see in similar literatures, e.g., on Bayesian methods. One option would be the uniform distribution, which however puts a lot of probability mass in the tails of least-cost and no adaptation. The most popular alternative, which puts more probability mass in the centre of the range, is the triangular distribution, and that is what we use.

We have expanded the description of our approach in the SI in order to try to make it clearer, more transparent, and better justified.

The study thus critically needs to address the above aspects, I otherwise do not find the damage estimates and the conclusions drawn from them worth presenting in Nature Comms.

A second, but less critical point is the necessity of the presented sea level model within the study. The study sticks to the standard RCP scenarios for which IPCC sea level numbers are available. SLR emulators like the one presented are most useful if you need to assess different scenarios than the standard ones. The economic assessment as such could be done with the official IPCC AR6 sea level estimates directly. It would be more straightforward and better for comparability to first use the study design with the standard IPCC numbers and maybe then explore additionally extreme scenarios like ABUMIP.

The reason for building a model of AIS melting is that otherwise we cannot calculate the social cost of carbon, which is a major outcome variable of interest in this study. The social cost of carbon is the economic cost of an incremental ton of CO₂ emissions. To calculate it, you need to run the coupled model twice, once on the baseline emissions scenario (i.e., the RCP), and then the second time with the extra ton of CO₂ on top of the baseline. Therefore, in effect we do need to assess different scenarios.

Minor comments:

title: "melting suggests the wrong process: it is mainly ice loss to the ocean"

This is a valid point. We have given a lot of thought to alternative titles. We concluded that melting is a useful concept to engage a wide audience (e.g., economists), while still staying true to the underlying processes as they involve either surface or basal melting. A quick google search indicates that "melting" is widely used in relation to Antarctic ice mass loss, including by reputed scientific institutions. However, to acknowledge this issue, we have added some text on lines 36-37: "we study the economic impacts of SLR from AIS mass loss, which we colloquially refer to as melting".

l24: "several dollars": needs relation to the total estimated cost of carbon

We agree and have rewritten the end of the abstract to be more precise and only discuss the relative increase in the social cost of carbon in percentage terms.

l26: more than what? total cost of carbon?

Yes, 50% more than the base social cost of carbon. We hope our attempt to redraft the text to address your previous minor comment makes this clear.

l41: unclear: storm surge or permanent inundation are not residual damages

What we mean by residual damage is costs of SLR after adaptation, which is reasonably common terminology in climate economics but perhaps not beyond. To try to clarify this, we have made two changes to the paper. We have simplified the text on line 41 so that it now simply says "costs broken down into different categories". This is because we want to keep the introduction short. Then, when we introduce CIAM in the section "Modelling approach in brief", we expand upon which cost categories we classify as adaptation costs and which we classify as residual damages. Please see lines 120-122.

l83: mass balance change?

Mass balance in glaciology does in fact refer to the change, so this is not a typo. Here is the definition of mass balance from the Scientific Committee on Antarctic Research: "The mass balance of a glacier or ice sheet is the net balance between the mass gained by snow deposition, and the loss of mass by melting (either at the glacier surface or under the floating ice shelves or ice tongues) and calving (production of icebergs)."

l265: what are NPV costs?

NPV stands for net present value – we have now spelled it out upon first mention.

l312: now 1% discount rate, not 4%?

These discount rates are not the same. The 4% discount rate used to discount monetary damages is known as the consumption discount rate. It is the rate at which to discount flows of money. The 1% figure refers to the pure rate of time preference, also known as the utility discount rate, because it is the discount rate applied to flows of utility, i.e., before conversion into money units. According to the standard Ramsey rule in welfare economics, the consumption discount rate $r = \rho + \eta g$, where ρ is the pure rate of time preference, η is the elasticity of marginal utility (which does the utility \rightarrow money conversion) and g is the growth rate of consumption per capita. If the economy is growing ($g > 0$), the second element is positive, and the consumption discount rate is larger than the pure rate of time preference. We choose $\eta = 1.5$, a fairly standard value in the literature. With $g \approx 2\%$ globally on aggregate (to calculate the social cost of carbon we do this in the conceptually correct way using country-specific growth rates that correspond to the SSP scenarios, so g is not an

assumption/primitive), $r \approx 4\%$. We have clarified in the text that the 4% rate is the consumption discount rate.

l338: conclusion also true already from DIVA 2014

This is quite true. We have added a sentence to clarify that “This result is consistent with previous studies looking at SLR from all sources.”

l452-l458: exactly: impacts are not easily separable per SLR contribution, but you do this for AIS in your main analysis.

Indeed, at the heart of the research design is our attempt to solve the issue that the impacts of different SLR contributions are not necessarily additive. By taking a modular approach so that different sources can, in effect, be switched on and off, we can account for any non-linearities. We are unsure whether this comment proposes a change to the text but guess not. Please tell us if we have misunderstood.

l463: your numbers thus depend also on all other components except the system is linear. therefore the numbers for total SLR should be given.

We agree with this comment and indeed we report total SLR in Figure 2. Please tell us if we have misunderstood what you are asking for.

l468: impact on consumption per capita? this is a first time mention. before, the reader thinks it is all about damages and adaptation direct costs.

The analysis of damages and adaptation costs can in principle be undertaken without estimates of consumption per capita. However, estimates of consumption per capita are essential when calculating the social cost of carbon. Your comment highlights that we failed to explain this clearly. We have therefore redrafted the section of the methods on “Estimating the social cost of carbon” to provide more clarity on this. We write:

“The incremental contribution of AIS melting to the social cost of carbon – the social cost of AIS melting – is calculated by running the META IAM with and without a contribution to SLR from AIS melting, estimating the social cost of carbon in both cases, and then taking the difference. The social cost of carbon is the present value of the economic costs of emitting an extra tonne of carbon dioxide. It is formally the monetary equivalent of the loss in global social welfare from the extra tonne of CO₂.⁴⁹ Global social welfare is standardly calculated as the discounted sum of national population times national utility per capita. National utility per capita is calculated by applying a utility function to national consumption per capita, net of the costs of climate change. See ¹⁴ for a full description of the welfare calculations in META.”

l S205: Country-level damages from SLR: good idea, but this completely relies on CIAM. There is a two order magnitude of adaptation vs no adaptation. is interpolation credible?

Our approach is to treat CIAM’s no-adaptation and least-cost-adaptation scenarios as upper and lower bounds respectively on country costs. We expect the reality to be somewhere in between – barriers to adaptation are well-documented (as mentioned in the paper), yet it would be unrealistic to expect countries not to adapt whatsoever, given how much costs can be reduced by doing so. This is why we interpolate, in this sense. Respectfully, we think it is a good approach to capture the uncertainty that is attributable to the adaptive response. More broadly though, you are of course quite right that our estimates are subject to the limitations of CIAM. Various kinds of misestimation

could result in costs outside these bounds. Please refer to our response to your major comment on this above.

Reviewer #3

The manuscript assesses the contribution of the melting of the Antarctic Ice Sheet (AIS) with a set of models tailored for the analysis in an integrated assessment framework. The authors build two reduced-forms models to describe the AIS behavior using data and information from recent literature. These two models are then coupled with the CIAM module integrated in the META (IAM) model to provide the economic cost of the AIS melting by comparing impacts from all sources of SLR with impacts from all sources of SLR less AIS melting.

Although the subject has already been studied in the past, as acknowledged in the manuscript, the differences highlighted by the authors are a contribution of new knowledge to the economic literature. Results show the importance of AIS in economic costs growing by the end of the century and that the AIS melting could increase the social cost of carbon from 7% to 53.3% on average depending on the scenario considered.

While there are some of points I think could be clarified and discussed in more detail in the manuscript, I recommend its publication after they have been addressed by the authors.

1. The fact that annual flow costs in 2100 are higher on RCP4.5 than on RCP8.5. Although the authors indicate that sequencing would be one of the main reasons explaining that result, it is not clear enough why costs in RCP 8.5 are decreasing after 2090, despite SLR is increasing even more. Could it be the negative contribution of SMB to SLR as shown in figure 2 or is it due to a reduction of population/asset exposure to SLR? Are there other reasons explaining that behavior?

Thank you for your comment – this is worth clarifying. Costs are only decreasing *temporarily* on RCP8.5, i.e., it is not a long-term phenomenon. The reason, as mentioned in the paper, is sequencing: faster SLR due to the extra contribution from the AIS can cause impacts to happen earlier. One indication that this isn't a long-term phenomenon is that the ordering across the three RCP scenarios of the discounted NPV of long-term costs is intuitive. Figure 9 in the SI shows that the discounted NPV of costs up to 2200 is ordered as you would expect: highest on RCP8.5, lowest on RCP2.6. To offer further evidence that cost decreases are temporary, here we show you the equivalent of Figure 3, bottom left panel, for the extended period to 2200. You can see the cost decrease in 2100 is a blip, with costs then increasing above the 2090 level in 2110 and beyond. You can also see the non-monotonicity of costs in the mid-2100s, although one should be careful not to overinterpret these uncertain, very long-term projections, which we don't directly use in the paper.

RCP8.5

2. It is not clear if the results shown for each RCP on the manuscript (figures 3-5) are in combination with a specific SSP or other socioeconomic data. Only in the SCC section there are RCP/SSP combinations. Please clarify this in the modelling approach section.

This is also worth clarifying, we fully agree. CIAM has its own, built-in socio-economic scenario, using the UN population projections and IMF income projections. These data are used for all our analysis of coastal impacts, also for calibrating the simplified, country-level coastal damage coefficients that go into META for the purposes of estimating the social cost of carbon (SCC). However, when META is run to estimate the SCC, the RCPs are combined with a specific SSP. Therefore, when we estimate the SCC, we need to assume that the potential mismatch between the CIAM socio-economic scenario, used to calibrate the SLR damage coefficients, and the SSPs does not seriously bias the results (controlling for SLR itself). This assumption is unavoidable on a practical level, because we cannot rewire the CIAM model within the scope of this paper (which makes income and population projections for each of the 12,148 DIVA segments), nor can we rewire the META model to accommodate the CIAM socio-economic scenario (of which there is only one in any case). We conducted some further analysis of this assumption. We compared country-level GDP growth rates in the CIAM socio-economic scenario with each of the SSPs. We calculated the average GDP growth rate between 2010 and 2100 for each country and below we report the average across countries of the relative difference in 2010-2100 growth rates between the CIAM scenario and each of the SSPs (a positive difference means growth is faster in the CIAM scenario). Overall, growth rates in the CIAM scenario are perhaps surprisingly close to SSP1 and SSP2 on average, albeit with significant variation in the difference, but growth is substantially slower on average in the CIAM scenario than in SSP5.

	Average difference	St. dev. of difference
SSP1	-4.7%	40.5%
SSP2	-2.8%	39.9%

SSP5	-36.6%	50.1%
------	--------	-------

We have added the following text to the SI: “Note this approach means the country cost coefficients derive from the CIAM socio-economic scenario, whereas META is based on the SSPs. CIAM’s socio-economic scenario has country growth rates close to SSP1 and SSP2 on average, but generally lower than SSP5.”

3. Line 297 mentions the combination RCP8.5-SSP2, but then lines 299-300 state that RCP8.5 should be paired with SSP5. I would encourage the authors to explain the choice of RCP8.5 with SSP2 and the implications for the SCC of an RCP8.5-SSP5 combination.

Apologies, this was a typo, it should have said RCP8.5-SSP5! We have corrected this.

4. Finally, the following reference could be included in the introduction:

Edwards, T.L., Nowicki, S., Marzeion, B. et al. Projected land ice contributions to twenty-first-century sea level rise. *Nature* 593, 74–82 (2021). <https://doi.org/10.1038/s41586-021-03302-y>

Thank you for pointing us to this paper, we evaluated where best to put a citation to it and eventually opted for p5, where we describe our SLR projections and compare them with IPCC AR6. Our understanding is that Edwards et al. adds value primarily by providing updated projections for the more recent SSP scenario framework and driven by CMIP6 global climate models.

REVIEWER COMMENTS

Reviewer #1 (Remarks to the Author):

This has been an overall quite responsive revision. The changes have improved the quality of this paper. There are just a few - in my view minor - loose ends that need to be fixed for this paper to be a high-quality contribution to the literature.

Line 303: the statement that all changes can be controlled for by the framework seems questionable. I would recommend to just delete this sentence and leave it at this.

Adding a survival function as suggested by review of 1 seems still to be a good idea in my assessment.

The response that the framework cannot be tested by hindcasts seems to be at odds with the current state-of-the-art. There are several studies that use hindcasts to assess and sometimes even calibrate this kind of model. For sure, the overall sea-level rise module can be tested and calibrated. Please add these details.

Please add even a short discussion of the assumptions that drive did the nature and dynamics of retreat. This issue is of considerable interest to the broad readership of this journal and the assumptions about retreat may be a key driver of the shown results.

Reviewer #2 (Remarks to the Author):

The authors address my raised points.

On the uncertainties from CIAM/DIVA, I however think they take it too easy. I would like like to have seen a clearer and more in-depth analysis here in particular on the interpolation between the adaptation / no adaptation extreme cases. Could you at least provide the reader some guidance where your interpolation typically lies in the range between the extreme cases? Do you through the triangle distribution just put most of the countries to "close to 0.5 adaptation"? Is this a reasonable choice?

Or is this more complex and you can see patterns or can you identify groups of countries that lie close to the no adaptation/high adaptation case? An illustration like Figure 4, but with the position in the range between no adaptation/adaptation extremes on the y axis may be an idea then. Nature Comms allows up to 10 displayed items. Then you may also discuss how this relates to what you see in Fig. 4.

You should also refer clearer in the main text to the interpolation between the extreme cases. So far this is only in the SI that not all readers will look at.

L450: Coastal adaptation and damage costs: clearer refer to the SI section on country level SLR damages.

Reviewer #3 (Remarks to the Author):

The authors have addressed all my comments. Thank you.

Economic impacts of melting of the Antarctic Ice Sheet

Response to Reviewers

9th August 2022

We would like to thank the two reviewers for their further comments. We have produced a revision that, we hope, addresses all the points raised. Below we provide a point-by-point response to the reviewers' comments. We would also like to flag that, upon making final quality checks of the model code and paper, we concluded that the description of the surface mass balance model on pages 1 and 2 of the SI needed some rewriting, in particular to improve the mathematical notation. We have done so. All changes to the paper have been tracked.

n.b. original comments in *italics*

Reviewer #1

This has been an overall quite responsive revision. The changes have improved the quality of this paper. There are just a few - in my view minor - loose ends that need to be fixed for this paper to be a high-quality contribution to the literature.

Line 303: the statement that all changes can be controlled for by the framework seems questionable. I would recommend to just delete this sentence and leave it at this.

We have done so.

Adding a survival function as suggested by review of 1 seems still to be a good idea in my assessment.

We thought about how to do this, given that the modelling is not based on survival analysis, so there isn't an unambiguous notion of survival. What we have opted to do is plot the probability that SLR from AIS melting stays below a series of threshold values (e.g., 0.1m) between 2010 and 2100. These threshold values correspond to the notion of survival. We do this for three threshold values (0.1m, 0.2m and 0.3m) and the three RCPs. These are plotted in Figure SI8.

The response that the framework cannot be tested by hindcasts seems to be at odds with the current state-of-the-art. There are several studies that use hindcasts to assess and sometimes even calibrate this kind of model. For sure, the overall sea-level rise module can be tested and calibrated. Please add these details.

This is fair. We have added a hindcast test of both the AIS melting model and the combined model of total SLR. This can be found in the SI, section 6, p14. We obtained observational data from IPCC AR6, focusing on the satellite altimetry era, 1993-2018. Consistent with Levermann et al. (2020), we find that the AIS melting model estimates higher SLR between 1993 and 2018 than the observations, although the model's 90% confidence interval is wide and envelopes the corresponding uncertainty about the observations. Conversely, estimated total SLR is lower than the observations, implying that the contributions from thermal expansion, glaciers and small ice sheets, and melting of the Greenland Ice Sheet in META are underestimated relative to the observations. Again, the 90% confidence intervals overlap.

Please add even a short discussion of the assumptions that drive did the nature and dynamics of retreat. This issue is of considerable interest to the broad readership of this journal and the assumptions about retreat may be a key driver of the shown results.

We have added a discussion at the start of the section “Coastal impacts”, which attempts to describe how adaptation happens in CIAM in a bit more detail, but still focusing on the basic intuition. We have also added two longer paragraphs to the SI starting at line 197, which describes in some detail what protection and retreat entail, and where the costs come from. We hope these additions are what you are looking for.

Reviewer #2

The authors address my raised points.

On the uncertainties from CIAM/DIVA, I however think they take it too easy. I would like to have seen a clearer and more in-depth analysis here in particular on the interpolation between the adaptation / no adaptation extreme cases. Could you at least provide the reader some guidance where your interpolation typically lies in the range between the extreme cases? Do you through the triangle distribution just put most of the countries to “close to 0.5 adaptation”? Is this a reasonable choice?

Or is this more complex and you can see patterns or can you identify groups of countries that lie close to the no adaptation/high adaptation case? An illustration like Figure 4, but with the position in the range between no adaptation/adaptation extremes on the y axis may be an idea then. Nature Comms allows up to 10 displayed items. Then you may also discuss how this relates to what you see in Fig. 4.

You should also refer clearer in the main text to the interpolation between the extreme cases. So far this is only in the SI that not all readers will look at.

These are fair points, thank you. What we have decided to do here is move the sentence discussing this issue from the methods section to the section on social cost of carbon results, and significantly expand it into a passage explaining more about how the country cost coefficients are calibrated and what is assumed about adaptation (see lines 295-303). Besides expanding the explanation in the main text, putting it here should allow the reader to intuitively relate the assumptions to the results in Figures 3-5, where the differences between the no adaptation scenario and the least-cost adaptation scenario are stark both at the global and country levels.

As we explain, we calibrate the upper and lower bounds of each country cost coefficient on the estimates of the full CIAM model under no adaptation and least-cost adaptation, respectively. We must then make an assumption about how the probability mass is distributed between these extremes, without having any solid data for calibration. We evaluated a number of options, noting that the adaptation literature strongly points towards adaptation happening, but imperfectly. One option is a uniform distribution, but in our opinion this overestimates the probability mass towards the extremes. Therefore, we prefer the triangular distribution, which is often used in Bayesian analyses to estimate non-uniform random parameters under limited information, and we further assume symmetry for want of any evidence on asymmetry. Therefore, your intuition is correct that, for any country in Figure 4, the modal adaptive response leads to costs halfway between what is estimated under no adaptation and least-cost adaptation.

L450: Coastal adaptation and damage costs: clearer refer to the SI section on country level SLR damages.

We have added a clear cross-reference to the SI in the new passage on how we estimated country-level SLR damages.

REVIEWERS' COMMENTS

Reviewer #1 (Remarks to the Author):

Thank you for the responsive and careful review. My comments and concerns have been addressed.

Reviewer #2 (Remarks to the Author):

The revised wording now more clearly states how damages are related to the original CIAM results. This is a minimal-effort revision, but ok with me at this stage.

One minor point: in L194 the authors write "Retreat is more likely to be chosen when the exposed land is of higher value."

I would expect the opposite, i.e. high value land is worth to protect, and retreat is not the likely option.

Economic impacts of melting of the Antarctic Ice Sheet

Response to Reviewers

13th September 2022

We have one remaining comment from Reviewer 2 to attend to:

One minor point: in L194 the authors write “Retreat is more likely to be chosen when the exposed land is of higher value.” I would expect the opposite, i.e. high value land is worth to protect, and retreat is not the likely option.

Apologies, the reviewer is correct, and this was a typo. We have switched “higher value” for “lower value”.